# Probiotics Attenuate Food Allergy via Short-Chain Fatty Acids-Mediated Immune Modulation and Gut Barrier Restoration

**DOI:** 10.3390/foods14223953

**Published:** 2025-11-18

**Authors:** Xue Feng, Liuying Li, Li Yan, Zhencong Yan, Zhoujin Xu, Yuting Fan, Philippe Madjirebaye, Xuli Wu

**Affiliations:** 1School of Public Health, Shenzhen University Medical School, Shenzhen University, Shenzhen 518060, China; 2300243013@email.szu.edu.cn (X.F.); liuying.li0724@gmail.com (L.L.); 2310246031@email.szu.edu.cn (L.Y.); 2310246004@email.szu.edu.cn (Z.Y.); 2310246025@email.szu.edu.cn (Z.X.); fanyuting@szu.edu.cn (Y.F.); 2Department of General Practice Medicine, Third Affiliated Hospital of Shenzhen University and State Key Laboratory of Respiratory Disease, Allergy Division, Shenzhen University, Shenzhen 518060, China

**Keywords:** probiotics, food allergy, short-chain fatty acids, regulatory T cells (Tregs), gut microbiota, intestinal barrier

## Abstract

The rising global prevalence of food allergy (FA) necessitates innovative therapeutic strategies. This study investigates the protective effects of three probiotic strains, *Lacticaseibacillus rhamnosus* HN001 (HN001), *Bifidobacterium lactis* HN019 (HN019), and *Lactobacillus acidophilus* NCFM (NCFM) against FA in a murine model. Probiotic administration significantly alleviated allergic symptoms and suppressed the Th2 response, reducing IgE, histamine, and cytokines (TNF-α, IL-2/5), while concurrently enhancing CD4+CD25+ regulatory T cell (Treg) activity and TGF-β1 expression. Treatment also restored intestinal integrity by upregulating tight junction proteins (ZO-1, claudin-1). 16S rRNA sequencing revealed that protection was underpinned by microbiota remodeling, marked by increased α-diversity and enrichment of SCFA-producing taxa (*Lachnospiraceae* and *Muribaculaceae*), which correlated with elevated acetate, butyrate, and propionate levels. Spearman analysis linked these microbial shifts to improved immune and barrier markers. Collectively, these findings demonstrate that probiotics mitigate FA through a convergent mechanism of immune rebalancing, barrier reinforcement, and SCFA-mediated microbiota-immune crosstalk, offering a promising microbiome-targeted therapy.

## 1. Introduction

Food allergy (FA) has emerged as a significant global health concern, substantially impairing the quality of life for affected individuals [1]. This immunological hypersensitivity reaction, predominantly triggered by the “big eight” allergenic foods (milk, eggs, peanuts, soybeans, tree nuts, wheat, shellfish, and fish) [2], arises from disrupted Th1/Th2 immune homeostasis. The pathogenic cascade begins when dendritic cells present food antigens to CD4+ T cells, preferentially activating Th2 lymphocytes that secrete IL-4 and IL-13 [3]. These cytokines drive B cell differentiation into IgE-producing plasma cells, while Th17-derived IL-17 exacerbates inflammatory responses [4]. The resulting elevation of sIgE, Th2-associated cytokines, and Th17-mediated inflammation collectively promotes intestinal inflammatory infiltration, ultimately manifesting as clinical FA symptoms [5,6].

Currently, no fully effective therapeutic alternatives exist for managing food allergies (FAs). However, emerging research highlights the critical role of probiotics in modulating host-microbiome interactions to alleviate FA. Advances in molecular technologies, particularly 16S rRNA gene sequencing and microbial community analysis, have revolutionized our understanding of how specific bacterial taxa influence the pathogenesis of FA. Evidence demonstrates that probiotics mitigate FA through multiple mechanisms, including the suppression of allergen-specific IgE (sIgE) and inflammatory cytokines, restoration of the Th1/Th2 balance, enhancement of intestinal barrier function, and modulation of gut microbiota composition [7]. Notably, specific probiotic strains exhibit broad anti-inflammatory effects against allergic conditions through gut microbiota remodeling and regulation of short-chain fatty acid (SCFA) metabolism [8]. SCFAs play a crucial role in maintaining intestinal homeostasis and regulating immune responses by recognizing receptors and inhibiting histone deacetylases, and are key molecules involved in the development and progression of allergic diseases [9,10]. Despite these promising findings, the precise molecular mechanisms underlying probiotic-mediated FA alleviation remain incompletely characterized.

This study demonstrates the anti-allergic effects of specific probiotics (*Lacticaseibacillus rhamnosus* HN001, *Bifidobacterium lactis* HN019, and *Lactobacillus acidophilus* NCFM) in a mouse model of food allergy. Probiotic administration, both as single strains and combinations, significantly alleviated clinical symptoms and suppressed key allergic mechanisms. This included rebalancing the Th2 immune response, evidenced by reduced IgE, histamine, and Th2 cytokines, and enhancing regulatory pathways via TGF-β1. Concurrently, probiotics restored gut barrier integrity and reshaped the gut microbiota, enriching it with taxa that produce short-chain fatty acids (SCFAs). The correlated rise in SCFAs provides a mechanistic link between probiotic-induced microbial shifts and systemic immunomodulation. While previous research has established the broad potential of probiotics in FA, many studies focus on a single mechanism or a single probiotic strain [11]. Our work advances this field by employing a multi-strain consortium and employing an integrated analytical approach to simultaneously map the immune, barrier, microbial, and metabolic consequences of intervention. This strategy allows us to delineate a cohesive, multi-faceted mechanism of action and assess potential synergistic effects between well-characterized strains, moving beyond associations to define a functional, SCFA-mediated pathway for FA alleviation [12]. These comprehensive findings underscore the potential of these probiotic strains as effective interventions for managing food allergies.

## 2. Materials and Methods

### 2.1. Materials

The study utilized the following commercial reagents: ovalbumin (OVA, ≥98% purity) from Sigma-Aldrich (St. Louis, MO, USA); HRP-conjugated goat anti-mouse IgE, IgG, and IgG1 (Fc-specific) antibodies from Southern Biotech (Birmingham, AL, USA); histamine and cytokine standards (TNF-α, IL-2, IFN-γ, IL-4, IL-5) from Jianglai Biotechnology (Shanghai, China); TGF-β1 and MCP-1 from Elabscience Biotechnology (Wuhan, China); and fluorescent-labeled antibodies (FITC anti-CD4, APC anti-CD25) from BioLegend (San Diego, CA, USA).

The study utilized 42 female BALB/c mice (6–8 weeks old, specific pathogen-free) sourced from the Guangdong Provincial Laboratory Animal Center (Guangzhou, China). All experimental procedures were approved by the Shenzhen University Animal Ethics Committee (Approval No. AEWC-202200007) and conducted in compliance with institutional guidelines. Mice were maintained under controlled environmental conditions (temperature, 24 ± 2 °C; humidity, 50 ± 10%) with a 12 h light/dark cycle in specific pathogen- and allergen-free facilities. Throughout the study period, mice had ad libitum access to food and water. After a 7-day acclimatization period, experimental procedures were initiated.

### 2.2. Probiotic Culture Conditions and OVA Solution Preparations

The probiotic strains used in this study, *Lacticaseibacillus rhamnosus* HN001 (batch number: 1103841178), *Bifidobacterium lactis* HN019 (batch number: 1103367072), and *Lactobacillus acidophilus* NCFM (batch number: 1103844189), were obtained from DuPont Nutrition & Biosciences (USA). These probiotics are commercially available, well-characterized probiotics have established health benefits, including immune modulation and gut microbiota regulation, as documented in previous studies [13,14,15,16,17].

For administration, each strain was resuspended in sterile phosphate-buffered saline (PBS, pH 7.4) to a final concentration of 10^9^ CFU/mL^−1^ [18,19,20]. Fresh suspensions were prepared daily to ensure optimal bacterial viability prior to oral gavage.

*Sensitization Solution Preparation*: Ovalbumin (OVA) was dissolved in physiological saline to prepare a 10 mg/mL (*w*/*v*) solution. This OVA solution was then mixed with physiological saline and aluminium adjuvant at a volume ratio of 14:5:1 (saline:adjuvant: OVA). The sensitization solution was prepared fresh for immediate use.

*Challenge Solution Preparation:* The OVA was dissolved in phosphate-buffered saline (PBS) to prepare a 150 mg/mL (*w*/*v*) solution. The challenge solution was prepared fresh for each use.

### 2.3. Construction of OVA-Induced Food Allergy Mouse Model

After one week of acclimatization, the mice were randomly allocated into six groups (7 mice in each group; Appendix A): negative control group (Control, physiological saline), food allergy group (Model, OVA only), *Lacticaseibacillus rhamnosus* HN001 group (HN001), *Bifidobacterium lactis* HN019 group (HN019), *Lactobacillus acidophilus* NCFM group (NCFM), and probiotic mixture group (Mix). All groups except the control group received oral OVA sensitization (200 μL) on days 0, 7, 14, and 21, while the treatment groups concurrently received daily probiotic suspensions (200 μL, 10^9^ CFU/mL). On day 28, after 12 h fasting, mice were challenged with high-dose OVA (200 μL). Allergic responses (symptom scores and diarrhea severity; Appendix A) were evaluated 30–60 min post-challenge. One hour post-stimulation, mice were euthanized for sample collection: blood from retro-orbital puncture for serum isolation and centrifugation at 3500× *g*, 15 min, 4 °C, spleen, duodenum (H&E staining), colon, and fecal samples. All samples were immediately flash-frozen (−80 °C) except duodenal tissues for histopathology.

### 2.4. Detection of OVA-Specific Antibodies in Mouse Serum

Previous methods were performed to detect specific IgE, IgG, and IgG1 antibodies using indirect ELISA in mouse serum [21,22]. Briefly, 96-well microplates were coated with 500 ng/well OVA in PBS (100 μL/well) and incubated overnight at 4 °C. After three washes with TBST (200 μL/well), non-specific binding was blocked with 5% BSA in PBS (200 μL/well, 37 °C, 1 h). Serum samples were diluted in blocking buffer (1:25; mg/mL, *w*/*v* for IgG/IgG1; 1:3; mg/mL, *w*/*v* for IgE) and incubated (100 μL/well, 37 °C, 2 h). After TBST washes, plates were incubated with HRP-conjugated secondary antibodies (100 μL/well, 1 h, 37 °C) and anti-mouse IgE, IgG, and IgG1. After final washes, color development was initiated with the TMB substrate (100 μL/well, 37 °C, 5–15 min in the dark) and stopped with 2 M H_2_SO_4_ (50 μL/well). Absorbance was measured at 450 nm using a microplate reader (Shenzhen Boxing Biotechnology Co.,Ltd., Shenzhen, China).

### 2.5. Quantification of Allergic Mediators and Cytokines by ELISA in Mice

According to the manufacturer’s instructions, the serum release levels of different allergic mediators (histamine and MCP-1) and cytokines (TNF-α, IL-2, IFN-γ, IL-4, IL-5, and TGF-β1) were detected using an ELISA kit with a multifunctional microplate reader (Shenzhen Boxing Biotechnology Co.,Ltd., Shenzhen, China).

### 2.6. Extraction and Analysis of Mouse Spleen Cells Related to Inflammatory Cytokines

The splenocyte isolation protocol was adapted from established methods [21]. Briefly, mouse spleens were gently homogenized through a 100 μm nylon mesh filter into RPMI-1640 medium. The resulting cell suspension was centrifuged (300× *g*, 5 min), and erythrocytes were lysed using 3 mL of ACK buffer. After PBS washing, viable splenocytes were resuspended in complete RPMI-1640 medium (2 mL/spleen). For cytokine analysis, cells (2.5 × 10^5^ cells per well) were cultured in 24-well plates with OVA stimulation (100 μg/mL) under standard conditions (37 °C, 5% CO_2_, 72 h). ELISA was used to analyze culture supernatants for Th1 (IFN-γ) and Th2 (IL-4, IL-5, IL-13) cytokine production, employing commercial kits according to the manufacturer’s protocols.

### 2.7. Detection of Treg Differentiation Rate in Mouse Spleen Cells

To assess Treg differentiation, splenocytes were adjusted to 2 × 10^6^ cells per well using the staining buffer. For flow cytometry analysis, cells were stained with fluorescent antibodies in separate tubes: a blank control (unstained), single-stained controls (FITC anti-mouse CD4 or APC anti-mouse CD25), and double-stained experimental samples. All tubes were incubated at 4 °C for 30 min in the dark, followed by centrifugation to remove supernatant. Cells were washed once with flow buffer, resuspended, and analyzed by flow cytometry [23,24].

### 2.8. Histopathological Analysis of Mouse Duodenal Tissue

Duodenal samples were fixed in 4% paraformaldehyde for 48 h and then processed for paraffin embedding. Tissue sections were stained using a hematoxylin and eosin (H&E) kit according to the manufacturer’s protocol. The sections were stained and resin-sealed and then observed under an inverted microscope.

### 2.9. Detection of Tight Junction Protein Expression in Mouse Colon

Proteins were extracted from colon tissue samples using radioimmunoprecipitation assay (RIPA) lysis buffer and quantified using a bicinchoninic acid (BCA) protein quantification Kit [25]. Subsequently, the proteins were heated at a constant temperature of 100 °C to denature them, and SDS-PAGE was used to separate them based on their different molecular weights. They then transferred to a polyvinylidene difluoride (PVDF) membrane and were blocked with 5% skim milk. After that, the membranes were soaked in the specific primary antibodies, including ZO-1, Claudin-1, and β-actin, overnight at 4 °C. The blocking membranes were then incubated in TBST (10 mmol/L Tris, 150 mmol/L NaCl, and 0.05% Tween-20, pH 7) to remove the unconjugated primary antibody, followed by incubation with the secondary antibody for 60 min while shaking gently. Finally, after washing, they were visualized using an enhanced chemiluminescence substrate (ECL). The Band intensity was quantified in Image J V2.1.4.7 software, using β-actin as the reference protein [26].

### 2.10. Changes in Gut Microbiota in Mouse Feces

The intestinal bacterial DNA was identified from the fresh feces of mice using a Mag-Bind Stool DNA Kit (Omega, Guangzhou, China) and confirmed by a NanoDrop NC2000 UV-vis spectrophotometer (Thermo Fisher Scientific Inc., Waltham, MA, USA). The PCR amplification was used to amplify the V3-V4 hypervariable region of the bacterial 16s rRNA gene, which was performed using universal primers (343F: 5′-TACGGRAGGCAGCAG-3′; 798R: 5′-AGGATCTAATCCT-3′). The amplicons were subjected to paired-end sequencing on the Illumina MiSeq platform of Personal Biotechnology, Co., Ltd. (Shanghai, China). The observed species, Chao 1, Shannon index, ACE, and principal coordinate analysis (PCoA) were used to evaluate the diversity. The taxonomic composition of the different groups was analyzed to identify the gut microbiota composition. The data was visualized using the Personalbio GenesCloud platform (https://www.genescloud.cn/, 25 June 2023) [27,28].

### 2.11. Short-Chain Fatty Acid Contents in Allergic Fecal Mice

Fecal short-chain fatty acids (SCFAs) were extracted by suspending samples in ice-cold 50% acetonitrile aqueous solution (containing internal standard), followed by vortexing, 20 min ice-bath incubation, and sonication. After centrifugation (12,000× *g* and 4 °C for 20 min), the supernatants were filtered through a 0.22 μm organic filter membrane and derivatized with 3-NPH and EDC-6% pyridine (in a 50% acetonitrile aqueous solution). The supernatant was then mixed with the derivatized solution and reacted at 40 °C for 30 min. Derivatized samples were recentrifuged and stored at −80 °C until UPLC-ESI-MS/MS analysis (Agilent, performed by Shenzhen Weinafe Biotechnology Co., Ltd., Shenzhen, China, 28 April 2023) for SCFA quantification.

### 2.12. Statistical Analysis

All experimental data were analyzed using GraphPad Prism 7 and are presented as the mean ± standard deviation. Multiple group comparisons were performed by one-way ANOVA, with statistical significance indicated as follows versus the model group: * *p* < 0.05, ** *p* < 0.01, *** *p* < 0.001, **** *p* < 0.0001; ns (not significant) *p* > 0.05.

## 3. Results

### 3.1. Probiotic Administration Attenuated Food Allergy Manifestations in Mice

Probiotic administration alleviated the severity and symptoms of allergy in mice, as assessed by food allergy (FA) symptom scoring. Initially, all groups exhibited similar symptoms (Figure 1A). After 30 days of OVA induction, the model group developed severe FA symptoms, confirming successful sensitization, which included significant weight loss. In contrast, probiotic-treated groups (*L. rhamnosus* HN001, *Bifidobacterium lactis* HN019, *Lactobacillus acidophilus* NCFM, and Mix) may mitigate weight reduction and markedly improve diarrheal symptoms (Figure 1B). Notably, the Mix group consistently showed the most pronounced improvement in clinical scores. However, this trend was not always statistically significant compared to the best-performing single strain, demonstrating strain-specific protective effects against FA.

### 3.2. Probiotic Treatment Modulated Allergic Antibody and Inflammatory Mediator Profiles

Probiotic treatment effectively modulated key immunological markers associated with food allergy in OVA-challenged mice. The model group exhibited significantly elevated serum levels of OVA-specific IgE, IgG1, and IgG, along with increased concentrations of the inflammatory mediators MCP-1 and histamine (*p* < 0.0001), confirming successful allergy induction (Figure 1C–G). Probiotic intervention substantially reduced these allergic response markers. To statistically assess potential synergistic effects, we directly compared the Mix group to the most effective single-strain group for each parameter. The Mix group demonstrated a significant enhancement in the suppression of histamine (*p* < 0.01) compared to all single-strain treatments (Figure 1G), indicating a synergistic interaction. For other markers like IgE, IgG1 and MCP-1, the Mix group’s effect was comparable to, but not significantly greater than, the best single strain. The downregulation of antibody responses and inflammatory mediators indicates that probiotics exert remarkable protective effects against FA through multiple immunological pathways.

### 3.3. Probiotics Alleviated Allergic Responses via Immune Modulation

Probiotic administration exerted notable immunomodulatory effects in allergic mice, demonstrating strain-specific capacity to rebalance both cytokine networks and regulatory T cell (Treg) responses. As evidenced in Figure 2A–F, probiotic intervention significantly attenuated pro-inflammatory TNF-α production, with *Bifidobacterium lactis* HN019 showing particularly potent suppression. The treatment simultaneously modulated Th1/Th2 polarization, effectively reducing elevated Th1-associated IL-2 while preferentially suppressing Th2-driven cytokines (IL-4 and IL-5), an effect most pronounced with the *Bifidobacterium lactis* HN019 and *Lactobacillus acidophilus* NCFM strains, though all treatments except *L. rhamnosus* HN001 and Mix showed significant IL-5 reduction. Notably, probiotics enhanced immune tolerance mechanisms through a marked upregulation of TGF-β1 expression and considerable expansion of CD4^+^CD25^+^ Treg populations (Figure 3). Statistical comparison revealed that the Mix group induced a significantly higher frequency of Tregs than any single-strain group (*p* < 0.05 versus *L. rhamnosus* HN001 and *Bifidobacterium lactis* HN019, the next best groups; Figure 3), providing clear evidence of a synergistic immunomodulatory effect. The upregulation of TGF-β1 in the Mix group was comparable to, but not significantly greater than of the three single-strain groups. These systemic immunomodulatory properties were further confirmed by cytokine modulation patterns observed in splenic immune markers (Figure 2G–I). Collectively, these findings establish that probiotics alleviate food allergy via suppressing Th2 responses while actively promoting Treg-mediated tolerance, effectively rebalancing the immune responses.

### 3.4. Probiotics Restore Duodenal Integrity and Enhance Tight Junction Proteins in Allergic Mice

Probiotics demonstrated protective effects on duodenal structure and intestinal barrier function in OVA-sensitized mice. Histological analysis (Figure 4A) revealed that the model group exhibited severe intestinal damage, including villus shedding, disorganized tissue structure, inflammatory infiltration, and lamina propria degradation. In contrast, probiotic-treated groups (*L. rhamnosus* HN001, *Lactobacillus acidophilus* NCFM, *Bifidobacterium lactis* HN019, and Mix) effectively reversed the severe duodenal damage observed in model mice, including villus atrophy, architectural disruption, inflammatory infiltration, and lamina propria degradation, with notable preservation of intestinal morphology.

Tight junction (TJ) proteins, such as ZO-1 and Claudin-1, play a crucial role in maintaining epithelial barrier function by preventing the penetration of allergens and pathogens [29]. As shown in Figure 4B, the model group displayed markedly reduced TJ protein expression, indicating increased intestinal permeability. However, probiotic supplementation, particularly with *Bifidobacterium lactis* HN019 and *Lactobacillus acidophilus* NCFM, exhibited robust effects in upregulating these barrier proteins (ZO-1 and Claudin-1, Figure 4C,D), reinforcing the intestinal barrier and promoting mucosal homeostasis. These findings suggest that probiotics mitigate food allergy-related damage by restoring gut structure and strengthening epithelial defense mechanisms.

### 3.5. Probiotics Restore Gut Microbiota Homeostasis in OVA-Induced Food Allergy Mice

The gut microbiota composition was analyzed via 16S rDNA sequencing of fecal samples from allergic mice. Probiotic treatment significantly enhanced α-diversity indices (Chao1, ACE, Shannon) compared to the model group (Figure 5A–C), demonstrating restoration of microbial richness and diversity despite OVA-induced dysbiosis. β-diversity analysis (Bray–Curtis PCoA, Figure 5D) revealed distinct clustering between the control and model groups, with all probiotic groups exhibiting microbiota profiles more similar to those of the control group, indicating partial normalization of community structure.

Probiotic treatment significantly modulated gut microbiota composition in allergic mice, as revealed by 16S rRNA taxonomic analysis. At the phylum level (Figure 5A), the relative abundance of Firmicutes increased in the model group. Bacteroidetes predominated across all probiotic groups (Figure 6B,C), significantly reducing the elevated Firmicutes/Bacteroidetes ratio observed in the model group (*p* < 0.001, Figure 6D). The model group showed enrichment of potentially harmful taxa, including *Desulfovibrionaceae*, while probiotic interventions promoted beneficial microbiota. *L. rhamnosus* HN001 and HN019 mice showed notable increases in *Muribaculaceae*, and *Bifidobacterium lactis* HN019 also enhanced *Lachnospiraceae* (Figure 6E–G). Heatmap analysis (Figure 6H) further demonstrated probiotic-specific modulation, with HN001 increasing the abundance of *GCA-900066575*, *Muribaculaceae*, *Muribaculum*, and *Prevotellaceae_UCG-001.* Probiotic *HN019* enriching *Alistipes*, *Lachnospiraceae_NK4A136_group*, *Prevotellaceae_UCG-001*, *ASF356* and *Rikenella*, and NCFM group promoting *Rikenellaceae_RC9_gut_group*, *Alistipes*, *UBA1819*, *Ruminococcaceae_UCG-009* and [*Eubacterium*]*_coprostanoligenes_group*. In contrast, *Clostridia_UCG-014*, *Desulfovibrio*, *Helicobacter*, *Escherichia-Shigella*, *Clostridia_vadinBB60_group*, and *Ruminococcus* were characterized in the model group. These findings suggest that probiotic strains differentially reshape gut microbial communities by suppressing pathogenic bacteria while enhancing beneficial taxa associated with the amelioration of FA.

### 3.6. Probiotics Modulate Gut Microbiota-Derived SCFAs in OVA-Sensitized Mice

Short-chain fatty acids (SCFAs) represent crucial intestinal metabolites that potentially influence the pathogenesis of food allergy. Our findings (Figure 7A–E) demonstrated significant increases in acetic acid, butyric acid, propionic acid, and pentanoic acid levels following probiotic administration, with *Lactobacillus acidophilus* NCFM, *L. rhamnosus* HN001, and *Bifidobacterium lactis* HN019 showing particularly pronounced effects. Notably, *Bifidobacterium lactis* HN019 and *Lactobacillus acidophilus* NCFM treatments also markedly elevated isobutyric acid concentrations compared to the model group. These results suggest that probiotic interventions may enhance SCFA production, potentially contributing to their protective effects against the development of FA.

### 3.7. Correlation Analysis Between Immune Markers, Gut Microbiota, and SCFAs in Allergic Mice

Spearman correlation analysis revealed significant relationships between gut microbiota, SCFAs, and immune markers in mice treated with probiotics (Figure 7F, Appendix A). Pathogenic genera (*Helicobacter*, *Acetatifactor*, *Desulfovibrio*, *Clostridia_UCG-014*, and *Clostridia_vadinBB60_group*) showed strong positive correlations with allergic mediators (IgE, IgG1, TNF-α, IFN-γ, IL-2/4/5, histamine, and MCP-1). Conversely, probiotic-associated taxa (*Lachnospiraceae_NK4A136_group*, *Lactobacillus*, *Blaudia*, *Lachnoclostridium*, *Rikenella*, *Anearotruncus*, *Alistipes*, and *Ruminococcaceae_UCG-009*) correlated positively with beneficial SCFAs (acetate, butyrate, propionate), regulatory immune factors (TGF-β1, CD4+CD25+ T cells), and tight junction proteins (ZO-1, Claudin-1). These findings suggest that probiotics alleviate food allergy by enriching SCFA-producing bacteria that modulate immune responses and enhance intestinal barrier function.

## 4. Discussion

Extensive research has established the potential of probiotics to modulate gut microbiota, regulate immune function, and mitigate inflammatory and allergic responses [30,31,32]. In the specific context of FA, this promise is linked to the restoration of microbial balance [33] and the promotion of immunomodulatory metabolites like SCFAs [34]. Building on this foundation, our study provides compelling evidence that a consortium of specific probiotic strains alleviates FA through a convergent mechanism involving immune rebalancing, gut barrier reinforcement, and the regulation of the microbiota-SCFA axis.

The significant alleviation of clinical allergic symptoms first evidenced the efficacy of our probiotic intervention. Crucially, we delineated the underlying immune mechanisms. A Th2-polarized response characterizes FA, driving IgE production and mast cell degranulation, which is marked by the release of histamine and MCP-1 [35,36,37,38,39]. Our findings demonstrate that probiotic administration effectively counteracts this state by suppressing key allergic effectors while enhancing immunoregulatory pathways. The observed induction of CD4+CD25+ regulatory T cells (Tregs) and TGF-β1 production is particularly significant, as this axis is critical for maintaining immune tolerance [40,41,42]. A remarkable restoration of gut barrier integrity underpinned this systemic immunomodulation. The probiotics conferred potent protection against OVA-induced intestinal damage, significantly restoring villus architecture and upregulating the tight junction proteins ZO-1 and Claudin-1. This confirms and extends previous work on the epithelial-protective effects of probiotics and butyrate [43,44,45], highlighting the restoration of the physical intestinal barrier as a fundamental mechanism in mitigating FA.

The primary driver of these systemic benefits appears to be a strategic reshaping of the gut ecosystem. Probiotic intervention enhanced microbial α-diversity, a hallmark of ecosystem health, and induced a selective enrichment of SCFA-producing taxa [46]. This included the enrichment of families like *Lachnospiraceae* and *Muribaculaceae*, renowned for their capacity to degrade dietary glycans and produce immunomodulatory metabolites [47,48,49], as well as other established SCFA-producers such as *Alistipes*, *Lactobacillus*, *Blautia*, and *Lachnospiraceae_NK4A136_group* [50,51,52]. This microbial restructuring was functionally consequential, correlating with a significant increase in cecal acetate, propionate, and butyrate.

These SCFAs are not merely biomarkers but active mediators of the observed protection. A key mechanism is their role as potent inhibitors of histone deacetylases (HDAC), which promotes the differentiation and function of Tregs via epigenetic reprogramming [53,54,55]. Our data empirically support and extend the recently proposed immune tone theory, which posits that high concentrations of microbial metabolites, such as SCFAs in the portal circulation, are crucial for educating immune cells and maintaining systemic immune homeostasis [56]. The increase in cecal SCFAs induced by our probiotic consortium provides a direct metabolic basis for this systemic immunomodulation, demonstrating a plausible pathway from gut microbiota remodeling to peripheral immune tolerance. The inhibition of HDAC increases histone acetylation, facilitates the transcription of genes critical for Treg lineage specification, most notably the Foxp3 gene, which is the master regulator of Treg development and function [57,58]. Our data, supported by the literature, suggest a direct mechanistic link where butyrate and propionate are particularly potent HDAC inhibitors (primarily of Class I HDACs), thereby promoting Treg differentiation and enhancing their immunosuppressive functions, such as IL-10 production [57,59]. This aligns with a foundational study demonstrating that SCFAs increase colonic Treg numbers and function via HDAC inhibition [55]. The relationship between gut microbiota and Treg differentiation is thus mediated by these bacterial metabolites, which act as a direct microbial signal to the host immune system. This SCFA-driven Treg induction is a crucial mechanism for maintaining intestinal homeostasis and suppressing aberrant inflammatory responses [53,60]. Furthermore, SCFAs also exert their effects through a complementary pathway by activating G-protein-coupled receptors (GPCRs), such as GPR43 (FFAR2) and GPR41 (FFAR3), which are expressed on immune cells and contribute to the overall immunomodulatory landscape [55,57,61]. The protection network was further reinforced by the concurrent suppression of pro-inflammatory genera implicated in allergic pathogenesis. Correlation network analysis effectively synthesizes these findings, revealing strong positive associations between probiotic-enriched taxa, SCFA levels, and improved clinical and immunological parameters. This integrated view demonstrates that the probiotics remodel the gut ecosystem towards a health-promoting state, whose metabolic output orchestrates a coordinated defense against FA through complementary pathways.

Despite these robust findings, certain limitations warrant consideration. First, the small sample size per group, although consistent with many preclinical studies [62,63], may limit statistical power for high-dimensional analyses such as microbiota diversity and correlation networks. While we observed statistically significant and biologically coherent patterns, a larger sample would better capture subtle microbial shifts and strengthen correlation analyses. Nonetheless, the substantial effect sizes in key metrics (such as SCFA levels and IgE reduction) and the consistency across immune, barrier, and metabolic endpoints support the validity of our conclusions. As a preclinical study, translation to human FA requires clinical validation. Additionally, while SCFAs were central mediators, future metabolomic analyses, including bile acids, could provide a more comprehensive view, and further work is needed to clarify strain-specific contributions and optimal dosing.

## 5. Conclusions

In conclusion, this study demonstrates that the probiotic consortium of *Lacticaseibacillus rhamnosus* HN001, *Bifidobacterium lactis* HN019, and *Lactobacillus acidophilus* NCFM alleviates FA via a convergent mechanism of immune regulation, barrier protection, and microbial restructuring. The probiotics attenuated the allergic response by rebalancing the Th2/Treg axis, suppressing IgE, histamine, and Th2 cytokines while enhancing Treg activity and TGF-β1. Concurrently, they restored intestinal barrier integrity by upregulating tight junction proteins (ZO-1, claudin-1). Crucially, these effects were underpinned by a reshaping of the gut microbiota, enriching SCFA-producing taxa and elevating acetate, propionate, and butyrate levels, which directly contribute to immune tolerance and barrier function. Based on our correlational data, we propose a model in which these SCFAs serve as central mediators, promoting immune tolerance and barrier function. Our findings position this probiotic formulation as a promising multi-targeted therapeutic strategy for FA.

## Figures and Tables

**Figure 1 foods-14-03953-f001:**
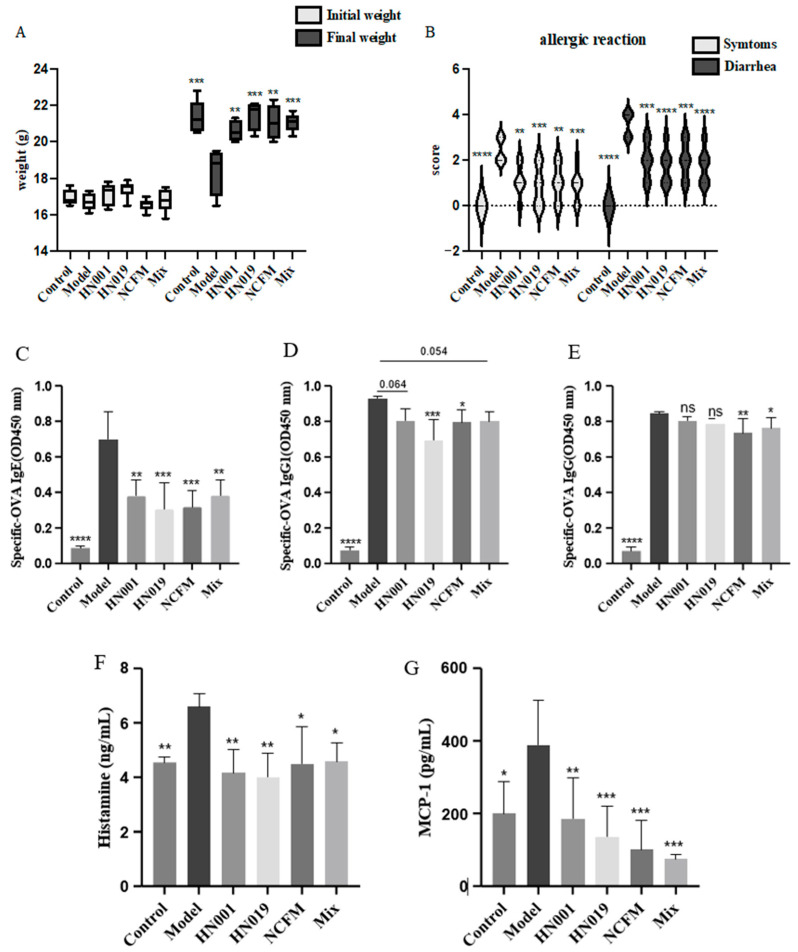
Probiotic supplementation alleviates allergic responses in mice. (**A**) Initial and final body weight of mice; (**B**) Allergic reaction score of mice; (**C**) OVA-specific IgE antibody; (**D**) OVA-specific IgG1 antibody; (**E**) OVA-specific IgG antibody; (**F**) histamine content; (**G**) The content of MCP-1. Data are presented as mean ± SD. * *p* < 0.05, ** *p* < 0.01, *** *p* < 0.001, **** *p* < 0.0001, ns: *p* > 0.05 compared with the model group.

**Figure 2 foods-14-03953-f002:**
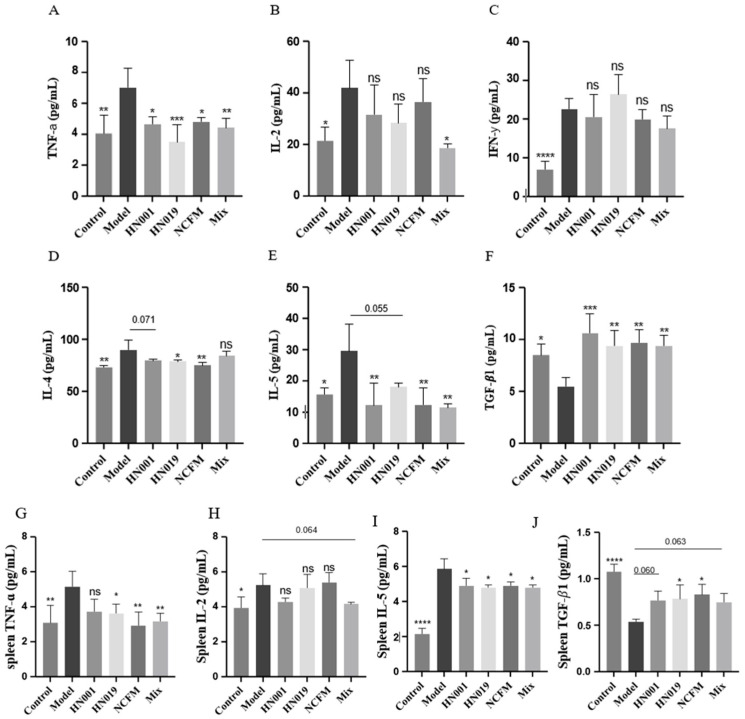
Probiotics modulate systemic and splenic cytokine profiles in OVA-induced allergic mice. (**A**–**F**) Serum levels of (**A**) TNF-α, (**B**) IL-2, (**C**) IFN-γ, (**D**) IL-4, (**E**) IL-5, and (**F**) TGF-β1. (**G**–**J**) Cytokine levels in splenic cell supernatants: (**G**) TNF-α, (**H**) IL-2, (**I**) IL-5, and (**J**) TGF-β1. Data represent mean ± SD. * *p* < 0.05, ** *p* < 0.01, *** *p* < 0.001, **** *p* < 0.0001, ns: *p* > 0.05 compared with the model group.

**Figure 3 foods-14-03953-f003:**
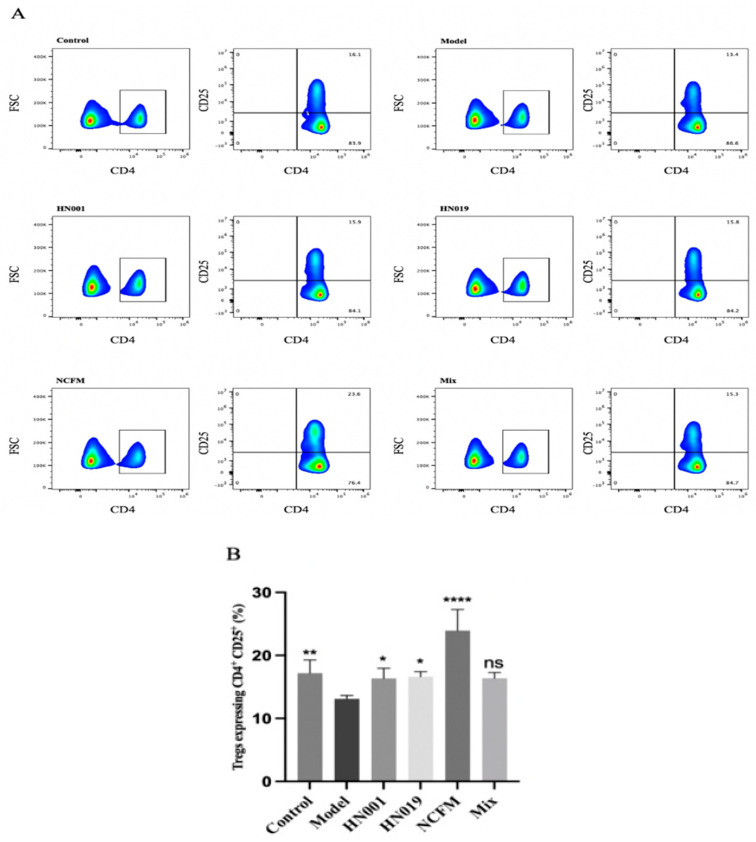
Probiotics enhance regulatory T cell (Treg) populations in OVA-induced allergic mice. (**A**) Flow cytometry gating strategy for CD4+ CD25+T cells. The boxed area in (**A**) highlights CD4+ T cells, with green indicating regions of high cell density and blue showing areas of relatively low density. (**B**) Frequency rate of CD4^+^ CD25^+^T cells. Data represent mean ± SD. * *p* < 0.05, ** *p* < 0.01, **** *p* < 0.0001, ns: *p* > 0.05 compared with the model group.

**Figure 4 foods-14-03953-f004:**
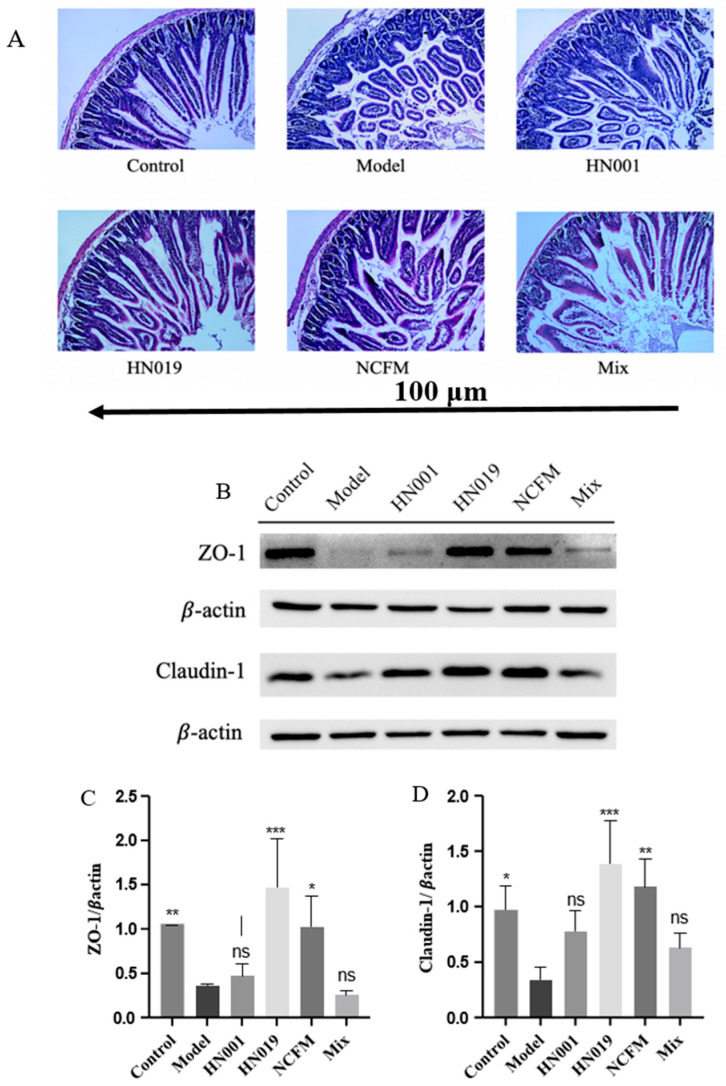
Probiotics restore intestinal integrity in OVA-induced allergic mice. (**A**) Representative H&E-stained duodenal sections (100 μm). (**B**) Probiotic modulation of tight junction protein signaling pathways, (**C**) Relative expression of ZO-1, and (**D**) Claudin-1 in duodenal tissue. Data represent mean ± SD. * *p* < 0.05, ** *p* < 0.01, *** *p* < 0.001, ns: *p* > 0.05 compared with the model group.

**Figure 5 foods-14-03953-f005:**
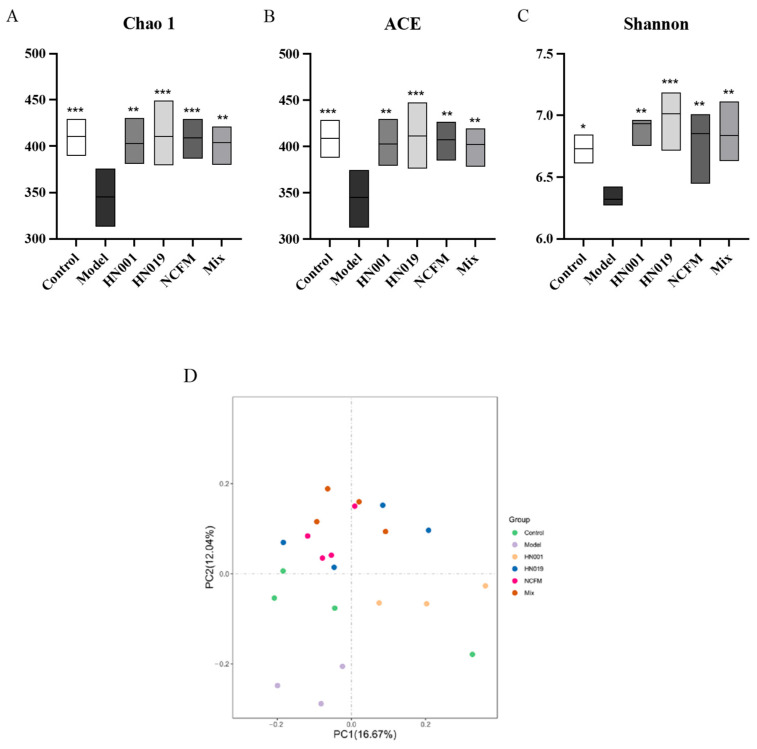
Probiotics modulate gut microbiota diversity in allergic mice. (**A**) Chao1 richness index; (**B**) ACE richness estimator; (**C**) Shannon diversity index; (**D**) Principal Coordinate Analysis (PCoA) of microbial communities (Bray–Curtis). Data are presented as mean ± SD. * *p* < 0.05, ** *p* < 0.01, *** *p* < 0.001 compared with the model group.

**Figure 6 foods-14-03953-f006:**
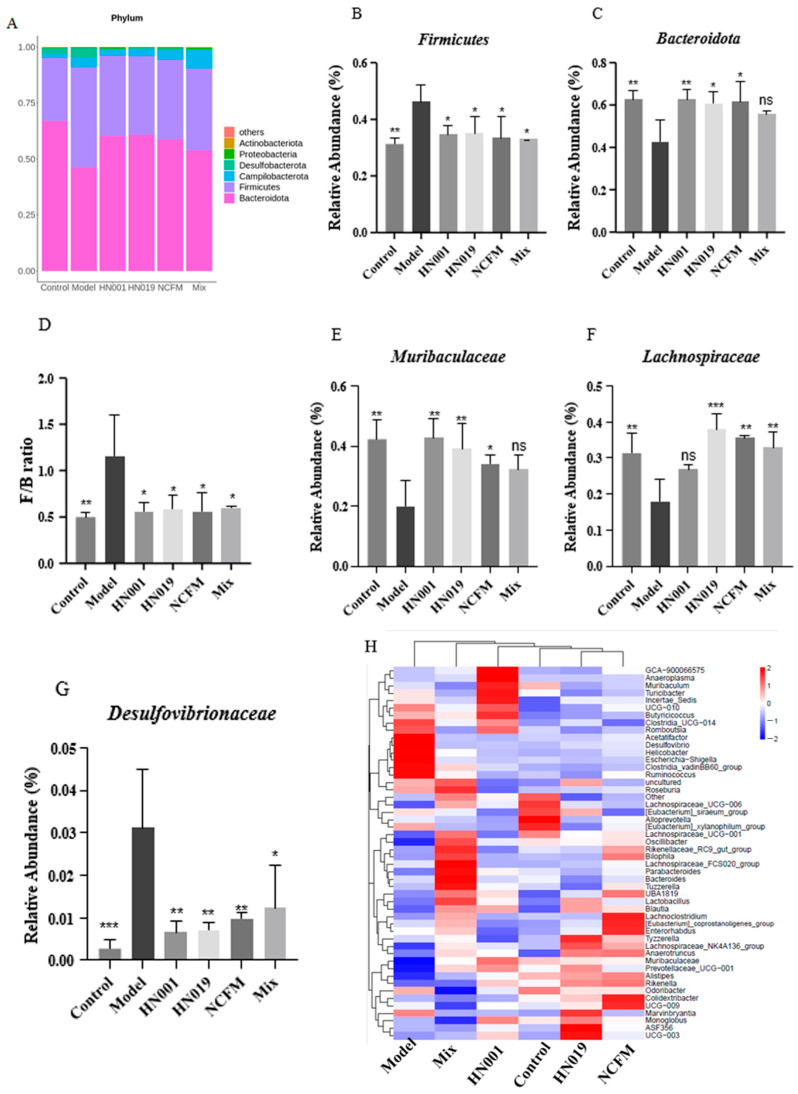
Probiotics reshape gut microbiota composition in OVA-allergic mice. (**A**) Phylum-level taxonomic profile; Relative abundance of Firmicutes (**B**) and Bacteroidetes (**C**); Firmicutes/Bacteroidetes (F/B) ratio (**D**); Relative abundance of (**E**) *Muribaculaceae*, (**F**) *Lachnospiraceae*, and (**G**) *Desulfovibrio*; (**H**) Heatmap of microbial composition (family level). Data represent mean ± SD. * *p* < 0.05, ** *p* < 0.01, *** *p* < 0.001, ns: *p* > 0.05 compared with the model group.

**Figure 7 foods-14-03953-f007:**
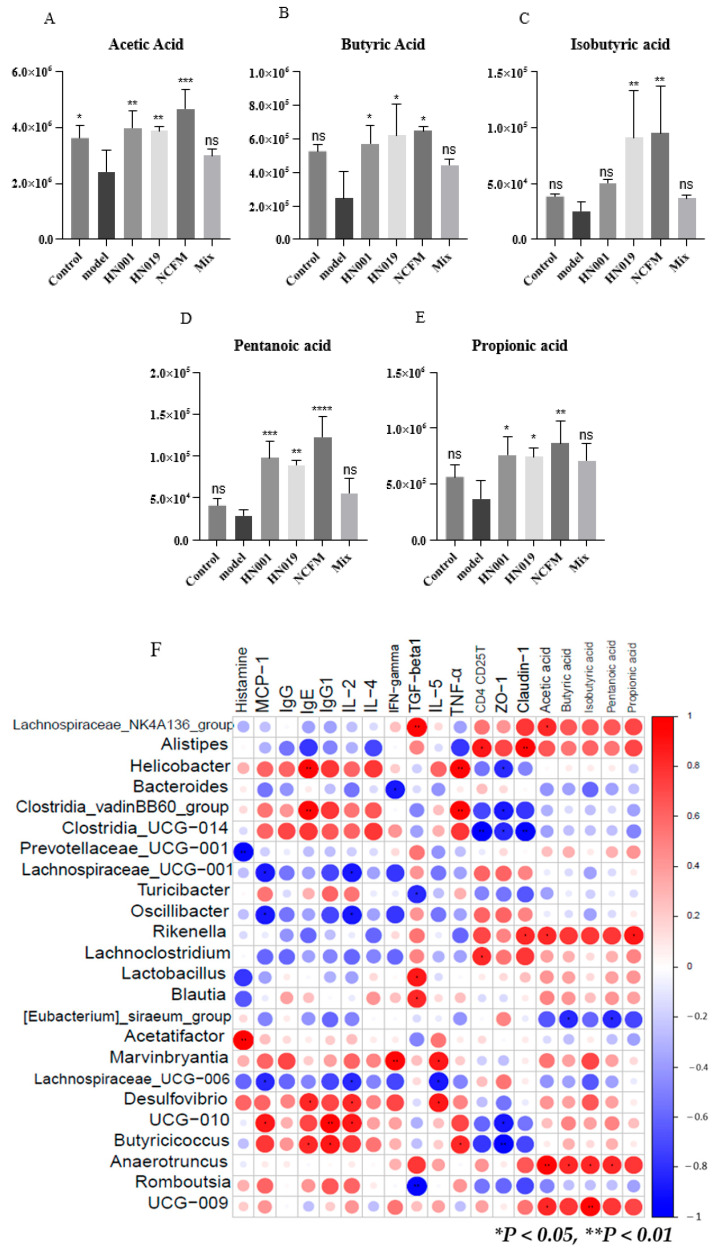
Probiotics enhance the production of short-chain fatty acids (SCFAs) and mediate microbial-host interactions in allergic mice. (**A**–**E**) Fecal SCFA concentrations: (**A**) acetate, (**B**) butyrate, (**C**) isobutyrate, (**D**) valerate, and (**E**) propionate. Data represent mean ± SD; * *p* < 0.05, ** *p* < 0.01, *** *p* < 0.001, **** *p* < 0.0001, ns: *p* > 0.05 compared with the model group. (**F**) Spearman correlation heatmap between gut microbiota (family level), SCFAs, and allergy parameters (|R| > 0.6; * *p* < 0.05, ** *p* < 0.01). Red indicates a high coefficient level, while blue indicates a low one.

## Data Availability

The original contributions presented in the study are included in the article and Appendix A, further inquiries can be directed to the corresponding authors.

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
