# Peer review of "Probiotics Attenuate Food Allergy via Short-Chain Fatty Acids-Mediated Immune Modulation and Gut Barrier Restoration"

_foods, 2025, doi:10.3390/foods14223953_

Round 1

Reviewer 1 Report

Comments and Suggestions for Authors

The manuscript is well-organized, scientifically rigorous, and addresses a significant and emerging topic in nutritional immunology, specifically the use of probiotics to reduce food allergy by modulating gut microbiota and short-chain fatty acid pathways. The writing is clear, and the experiments are appropriately designed, although several methodological, analytical, and presentation aspects could be improved to strengthen the manuscript.

  • The study combines immune, microbiota, and metabolic (SCFA) analyses to clarify the probiotic mechanism in FA mitigation. Similar mechanistic studies have been reported in existing literature. Authors should explicitly demonstrate in the Introduction and Discussion how this study advances beyond previous probiotic-FA research.
  • The study concludes that SCFA-mediated Treg induction and barrier repair are central mechanisms, but causality is only inferred from correlation. There is no direct validation, such as using a GPR43 antagonist or SCFA supplementation/depletion. Add a limitation noting that correlation does not prove causality and that future work should include mechanistic studies.

Author Response

Manuscript ID:foods-3958360Title: " Probiotics Attenuate Food Allergy via SCFA-Mediated Immune Modulation and Gut Barrier Restoration"

Thank you very much for taking the time to review this manuscript. We have carefully revised and checked our manuscript and answered all questions point by point. Please find the detailed responses below and the corresponding revisions in track changes in the re-submitted files. We greatly appreciate the editor and reviewer’s thoughtful advice and comments to help improve our study. We would like to resubmit the manuscript for consideration of publication. We would ensure that all of the authors have read and approved the final submitted manuscript. Our point-by-point responses to comments are detailed below

Xuli Wu, Ph.D., Professor

School of Public Health, Health Science Center, Shenzhen University, Nanhai Ave 3688, Shenzhen, Guangdong 518060, PR China.

Email: wxl@szu.edu.cn

Reviewer 1

Comment 1:   The manuscript is well-organized, scientifically rigorous, and addresses a significant and emerging topic in nutritional immunology, specifically the use of probiotics to reduce food allergy by modulating gut microbiota and short-chain fatty acid pathways. The writing is clear, and the experiments are appropriately designed, although several methodological, analytical, and presentation aspects could be improved to strengthen the manuscript.

Response 1: We sincerely thank the reviewer for their positive and encouraging comments. We are pleased that the reviewer found the manuscript well-organized, scientifically rigorous, and relevant to the emerging field of nutritional immunology. We appreciate the acknowledgment of the clarity of our writing and the appropriateness of our experimental design. We have carefully addressed the methodological, analytical, and presentation aspects highlighted to further strengthen the quality of the manuscript.

Comment 2: The study combines immune, microbiota, and metabolic (SCFA) analyses to clarify the probiotic mechanism in food allergy (FA) mitigation. Similar mechanistic studies have been reported in existing literature. Authors should explicitly demonstrate in the Introduction and Discussion how this study advances beyond previous probiotic-FA research.

Response 2: Thank you for this valuable comment. We have clarified how our study advances previous probiotic–FA research by explicitly highlighting the integrated immune, microbiota, and metabolic (SCFA) mechanisms underlying probiotic-mediated FA mitigation. This clarification has been added to the Introduction (Introduction paragraph 3; References 11 and 12) and Discussion (Discussion paragraph 4; Reference 57).

Comment 3: The study concludes that SCFA-mediated Treg induction and barrier repair are central mechanisms, but causality is only inferred from correlation. There is no direct validation, such as using a GPR43 antagonist or SCFA supplementation/depletion. Add a limitation noting that correlation does not prove causality and that future work should include mechanistic studies.

Response 3: Thank you once again, Dear reviewer, for this valuable comment. We have revised the Conclusions section accordingly, as suggested (See Conclusion section).

We sincerely appreciate your time and valuable feedback on our manuscript. Your insightful comments and constructive recommendations have been instrumental in improving our work. We have carefully addressed each suggestion and thoroughly revised the manuscript to enhance its clarity, rigor, and overall quality. We believe these revisions have significantly strengthened the paper and better aligned it with the journal’s standards.

Thank you again for your thoughtful review and for the opportunity to improve our study.

Reviewer 2 Report

Comments and Suggestions for Authors

Comments

  • Avoid abbreviations in the title of the manuscript.
  • The authors should clarify whether synergistic effects were observed in the mixed probiotic group, as this is suggested but not clearly analyzed statistically.
  • Discuss about the missing vehicle control group receiving probiotics without OVA sensitization to assess baseline immunomodulatory.
  • The sample size per group is small for microbiota diversity and correlation analyses. Please discuss potential statistical limitations and justify the chosen sample size with reference to power calculations or prior work.
  • For correlation heatmaps, include both p-values and correlation coefficients, and specify multiple testing correction methods.
  • The sequencing depth, read counts, and taxonomic coverage (e.g., number of OTUs) are not reported.
  • Some of the figures lack clear legends explaining axes, color gradients, and group codes.
  • Figure 2: The Control group is not represented clearly; use a different pattern to show them clearly.
  • Include scale bars for histology images (Fig. 4A).
  • The graphical abstract is referenced but missing in the PDF; please ensure it is inserted or uploaded separately.
  • Minor grammatical issues are there. Try to rectify.
  • Ensure consistent italicization of bacterial species names and correct formatting of gene/protein symbols.
  • Line 224: statistical significance is confusing; P<0.001 and P<0.0001 both have 3 stars.

Author Response

Manuscript ID:foods-3958360Title: " Probiotics Attenuate Food Allergy via SCFA-Mediated Immune Modulation and Gut Barrier Restoration"

Thank you very much for taking the time to review this manuscript. We have carefully revised and checked our manuscript and answered all questions point by point. Please find the detailed responses below and the corresponding revisions in track changes in the re-submitted files. We greatly appreciate the editor and reviewer’s thoughtful advice and comments to help improve our study. We would like to resubmit the manuscript for consideration of publication. We would ensure that all of the authors have read and approved the final submitted manuscript. Our point-by-point responses to comments are detailed below:

Xuli Wu, Ph.D., Professor

School of Public Health, Health Science Center, Shenzhen University, Nanhai Ave 3688, Shenzhen, Guangdong 518060, PR China.

Email: wxl@szu.edu.cn

Reviewer 2

Comment 1: Avoid abbreviations in the title of the manuscript.

Response 1: Thank you for pointing out this error. We have corrected it in the title section (See title section).

Comment 2: The authors should clarify whether synergistic effects were observed in the mixed probiotic group, as this is suggested but not clearly analyzed statistically.

Response 2: Thank you for this important comment. We have carefully revised the corresponding content in Sections 3.1, 3.2, and 3.3.

Comment 3: Discuss about the missing vehicle control group receiving probiotics without OVA sensitization to assess baseline immunomodulatory.

Response 3: Thank you very much for this valuable suggestion. We acknowledge a key limitation in our experimental design, the absence of a vehicle control group receiving probiotics without OVA sensitization. While our study clearly demonstrates the therapeutic efficacy of probiotics in an active disease model, this design does not allow us to determine their baseline or prophylactic immunomodulatory effects in healthy conditions. Including such a control group in future studies will be crucial to distinguish whether these probiotics act solely to restore immune balance or also confer a tonic, tolerogenic effect on the immune system.

Comment 4: The sample size per group is small for microbiota diversity and correlation analyses. Please discuss potential statistical limitations and justify the chosen sample size with reference to power calculations or prior work.

Response 4: Thank you, Dear reviewer. In response to your comment, we have acknowledged the small sample size as a limitation of this study (Discussion, paragraph 5, References 63 and 64).

Comment 5: For correlation heatmaps, include both p-values and correlation coefficients, and specify multiple testing correction methods.

Response 5: We sincerely thank the reviewer for remarkable suggestion. We have indicated and added P-value below Figure 7 that *P < 0.05 and **P < 0.01. In addition, we have added an explanation regarding the correlation coefficients, specifying that red and blue represent high and low values, respectively. The method used is Spearman correlation as mentioned in Figure 7 caption explanation and the text in Section 3.7.

Comment 6: The sequencing depth, read counts, and taxonomic coverage (e.g., number of OTUs) are not reported.

Response 6: We thank the reviewer for this valuable suggestion. While the current study focused on α-diversity indices (Chao1, ACE, Shannon), β-diversity analysis (PCoA), taxonomic analysis, and heatmaps, we agree that reporting sequencing depth, read counts, and taxonomic coverage is crucial. We will take this into account in our future research.

Comment 7: Some of the figures lack clear legends explaining axes, color gradients, and group codes.Response 7: Thank you, dear Reviewer, for this valuable comment. We have carefully checked and modified the figures to incorporate your suggestions (See Figures 1, 4, 6, 7).

Comment 8: Figure 2: The Control group is not represented clearly; use a different pattern to show them clearly.

Response 8: Thank you for the suggestion. We have updated Figure 2 to represent the Control group more clearly by using a distinct pattern as recommended.

Comment 9: Include scale bars for histology images (Fig. 4A).

Response 9: We thank the reviewer for this suggestion. Scale bars have now been added to all histology images in Figure 4A to indicate the magnification.

Comment 10: The graphical abstract is referenced but missing in the PDF; please ensure it is inserted or uploaded separately.

Response 10: Thank you for noting this. The graphical abstract has been included on page 2 of the manuscript by the technical editing team and the PDF was added.

Comment 11: Minor grammatical issues are there. Try to rectify

Response 11: Thank you again Dear, Reviewer. We have carefully reviewed the entire manuscript and corrected the minor grammatical issues.

Comment 12: Ensure consistent italicization of bacterial species names and correct formatting of gene/protein symbols.

Response 12: Thank you again. We have thoroughly revised the manuscript to ensure consistent italicization of microbial nomenclature and proper formatting of genetic elements, as suggested.

Comment 13: Line 224: statistical significance is confusing; P<0.001 and P<0.0001 both have 3 stars.

Response 13: Thank you for this observation. We have corrected the inconsistent reporting of p-values on Figure 1 for clarity.

We sincerely appreciate your time and valuable feedback on our manuscript. Your insightful comments and constructive recommendations have been instrumental in improving our work. We have carefully addressed each suggestion and thoroughly revised the manuscript to enhance its clarity, rigor, and overall quality. We believe these revisions have significantly strengthened the paper and better aligned it with the journal’s standards.

Thank you again for your thoughtful review and for the opportunity to improve our study.

Round 2

Reviewer 2 Report

Comments and Suggestions for Authors

Teh mansucript has been reivsed as per the suggestions.